

# The potential role of CD8+ cytotoxic T lymphocytes and one branch connected with tissue-resident memory in non-luminal breast cancer

Ziqi Zhao, Xinyu Ma and Zhengang Cai

Department of Breast Cancer, The First Affiliated Hospital of Dalian Medical University, Dalian, Liaoning Province, China

## ABSTRACT

Advances in understanding the pathological mechanisms of breast cancer have resulted in the emergence of novel therapeutic strategies. However, triple-negative breast cancer (TNBC), a molecular subtype of breast cancer with a poor prognosis, lacks classical and general therapeutic targets, hindering the clinical application of several therapies to breast cancer. As insights into the unique immunity and molecular mechanisms of TNBC have become more extensive, immunotherapy has gradually become a valuable complementary approach to classical radiotherapy and chemotherapy. CD8+ cells are significant actors in the tumor immunity cycle; thus, research on TNBC immunotherapy is increasingly focused in this direction. Recently, CD8+ tissue-resident memory (TRM) cells, a subpopulation of CD8+ cells, have been explored in relation to breast cancer and found to seemingly play an undeniably important role in tumor surveillance and lymphocytic infiltration. In this review, we summarize the recent advances in the mechanisms and relative targets of CD8+ T cells, and discuss the features and potential applications of CD8+ TRM cells in non-luminal breast cancer immunotherapy.

## INTRODUCTION

Early trends in breast cancer research focused on killing tumor cells directly, bringing radiotherapy and chemotherapy into general use. However, both of them do not localize adequately to the tumor site, and the accidental destruction of normal tissues and systems limits the clinical benefit of these techniques and even hinders a good prognosis. Thus, there is no doubt that the development of immunotherapy provided a tremendous infusion of hope to breast cancer patients with hormone receptor-positive and HER2+ breast cancer. Currently, chemotherapy remains the mainstream treatment method for triple-negative breast cancer (TNBC), which accounts for over 15–20% of breast cancer cases without the classical hormone receptor (HR) and HER2 (*Garrido-Castro, Lin & Polyak, 2019*). Whatever the choices of times for TNBC chemotherapy, metastases and relapses and drug resistances are more likely to be caused in these patients without proper

Corresponding authors
Xinyu Ma, maxinyu0401@163.com
Zhengang Cai,
caizhengang@firsthosp-dmu.com

maintenance therapy as they are throughout other type breast cancers. Moreover, the molecular heterogeneity within TNBC cells should also take on a fairly negative role which lead to a more aggressive disease course and low immune response underlie the poor prognosis of TNBC compared with other molecular types of breast cancer. Thus, the application value of immunotherapy in TNBC is revealed through the investigation of multiple targets.

Though there are several types of tumor-infiltrating cells, we mainly discuss the high levels of CD8+ cytotoxic T lymphocytes (CTLs) in breast cancer in this review. A previous systematic review including 13,914 cases of breast cancer confirmed that a median of about 60% and 61% of TNBC and HER2+ cases, respectively, contain CD8+ T cells, compared with only 43% of HR+ breast cancers (*Stanton, Adams & Disis, 2016*). TNBC and HER2+ breast cancer are considered to have high immunogenicity, which equates to their being slightly hotter from the viewpoint of immunity, although breast cancer in general tends to be cold (*Corti, Nicolò & Curigliano, 2021*). In addition to the total count of T cell, the density and localization of this kind of immune cell is also crucial to the therapeutic efficacy. Clinical data indicate that a high level of tumor-infiltrating lymphocytes (TILs) is aligned with favorable outcomes and response rates to chemotherapy in solid tumors, with the density of CTLs in the tumor being particularly important (*Koh et al., 2017*; *Nelson et al., 2021*; *Topalian et al., 2016*). These results further indicate that focusing more on CD8+ T cells has extensive prospects for the treatment of HR– breast cancer, especially for TNBC. Furthermore, we can conclude that the infiltration of CTLs is helpful for subsequent evaluations of breast cancer and even determine whether intensive adjuvant therapy or immunotherapy need to be plugged into clinical strategies.

## THE APPLICATION OF CD8+ T CELLS IN BREAST CANCER

The use of immune checkpoint inhibitors (ICBs) as immunotherapy has matured. Clinical research on breast cancer showed that the dominant blocking sites consist of CTL-associated antigen 4 (CTLA-4) and programmed death receptor 1 and ligand 1 (PD-1/PD-L1). In contrast to conventional immunotherapy, which also has a killing effect on tumor cells, ICBs play an indirect role in anti-tumor immunity by enhancing the immune response driven by CD8+ T cells and/or alleviating the depletion of CTLs (*Farhood, Najafi & Mortezaee, 2019*). CTLA-4 blockers, including ipilimumab, can strengthen the activity of CD4+ T cells, which affects CD8+ T cells. PD-1 and its coupled PD-L1 on the surface of tumor cells and PD-L2 attached to dendritic cells (DCs) and macrophages can decrease the production of IFN-γ *via* certain interactions and/or individually, and the associated checkpoint inhibitors mainly comprise nivolumab and pembrolizumab (MK-3475) (*Kansy et al., 2017*; *He et al., 2019*). Several Phase III clinical trials have already demonstrated significant benefits for overall survival (OS), progression-free survival (PFS), and pathological complete response (PCR) using anti-PD-1/PD-L1. Therefore, regular first-line chemotherapy and neo-adjuvant chemotherapy in combination with ICBs have been approved as the standard treatment regimens for PD-L1-positive TNBC (*Schmid et al., 2020b*; *Cortes et al., 2020*; *Schmid et al., 2020a*). However, one study showed that anti-PD-1/PD-L1 on its own might

lead to the upregulation of Tim-3, one of the checkpoints, thereby increasing the expression of PD-1 (*Tumeh et al., 2014*). Therefore, the blockage of two or even multiple targets is theoretically superior to antagonizing a single target, such as PD-1-Tim-3 blockade plus B and T cell lymphocyte attenuator (BTLA) blockade or PD-1 and/or CTLA-4 blockade plus a CD27 agonist (*Farhood, Najafi & Mortezaee, 2019*; *Van De Ven & Borst, 2015*).

Compared with the ICBs mentioned above, chimeric antigen receptor (CAR)-T cells, which fall into the category of adoptive cell therapy (ACT), have even more direct and immediate cytotoxic activity, especially in TNBC with high immunogenicity. The primary components and functional units of engineered T cells that have been constructed *in vitro* are enriched with extracellular single-chain variable fragments (scFvs) welded to the surface of the cells through the flexure hinge, transmembrane domain, and intracellular signaling domain, including the CD3 ζ-chain intracellular domain and co-stimulating domain. These kinds of constructs can release pro-inflammatory cytokines, promote the expression of the Fas ligand (FasL/CD95L/CD178) and TNF-related apoptosis-inducing ligand (TRAIL), and secrete cytotoxic granules containing perforin and granzymes to induce target cell apoptosis (*Xie et al., 2020*). Furthermore, genetically engineered T cells with the gene transfer of CAR *via* a viral vector also have the potential for proliferation, such as the CLASH system with adeno-associated viruses (*Corti et al., 2022*; *Dai et al., 2023*). The characteristics of CAR-T confer better stability, higher coverage, and greater specificity in comparison with other treatments (*Xie et al., 2020*). So far, specific targets reported in breast cancer include AXL, FRα, TROP2, EGFR (HER1), mesothelin (MSLN), SSEA-4, tumor endothelial marker (TEM8/ANTXR1), CSPG4 (MCSP/HMW-MAA), αvβ3, c-Met (HGFR), NKG2D ligand (NKG2DL), ICAM-1 (CD54), GD2, ROR1, MUC1, CD44v6, EpCAM, EphA10, Notch, CLDN18.2 and endosialin (CD248) (*Xie et al., 2020*; *Cao et al., 2022*; *Ash et al., 2024*). These can be involved in building scFvs, especially for TNBC. The targets mentioned above have been suggested to be correlated with the proliferation, migration, anti-apoptotic characteristics, invasion, and drug resistance of tumor cells, as well as tumor angiogenesis and T cell immunosuppressive effects (*Xie et al., 2020*; *Corti et al., 2022*). As to HER2+ breast cancer, CARHer2 T cell is possibly conductive to induce more Th1 cytokines which can kill more tumor cells (*Xie et al., 2023*). Even though the correlation between different targets and tumor cells and the feasibility of incorporating these targets into the construction of engineered CD8+ T cells have been demonstrated, there are challenges surrounding the application and development of CAR-T, such as the restriction of transportation and homing due to the suppressive tumor microenvironment (TME) and the limited motivation of the construction, immune escape caused by tumor heterogeneity, extracellular toxicities including cytokine release syndrome (CRS), and immune effector cell-associated neurotoxicity syndrome (ICANS) (*Corti, Nicolò & Curigliano, 2021*; *Ash et al., 2024*; *Shah et al., 2023*). It has been demonstrated that CAR-T cell treatment would have the better anti tumor effect combined with some kind of specific target therapy, such as PD-L1 and vascular endothelial growth factor (VEGF) antibody (*Gao et al., 2023*; *Dong et al., 2023*). To boost CD8+ T cells, it is supposed to improve the structure of CAR-T, for example, to activate T cells more

accurately and efficiently and modify the immune response *via* two or even multiple targets comprising different antigens. Currently, FRα plus MLSN, HER2 plus MUC1, and PSMA plus PSCA are undergoing preclinical studies (*Xie et al., 2020*). Moreover, engineered cells can be encoded by introducing suicide genes into T cells that regulate their exhaustion or screening specific targets that overlap minimally with normal tissues and/or maximally with suppressive immune cells (*To et al., 2022*), apart from addressing adverse effects by monitoring related clinical indicators passively to address extracellular toxicities (*Thompson et al., 2019*).

In addition, the T cell receptor (TCR) and co-stimulating signaling domain in CAR-T contribute to solving the problem to a large degree that the recognition and activation of T cells is influenced by MHC restriction. Based on the first-generation CAR-T modified by the CD3 ζ single chain, an extra co-stimulating domain has been added in second-generation CARs; these mature co-stimulating domains, such as 4-1BB (CD137) or CD28, play a vital role in the construction of various CAR-T. Compared with CD28, 4-1BB might better augment memory cell function as well as prolong the time of survival and activity (*Kawalekar et al., 2016*). Other co-stimulating domains in third-generation CARs include ICOS (CD278), OX40 (CD34), DAP10, and CD27 (*Corti et al., 2022*; *Honikel & Olejniczak, 2022*). The fourth-generation CARs can also activate downstream transcription factors to induce the production of non-specific cytokines and target tumor tissues rather than the regular enhancement of CD8+ T cells (*Chmielewski & Abken, 2015*), by which remodeling CTLs enhance cytotoxicity against tumor cells, in principle. Currently, CAR-T therapy for solid tumors, such as breast cancer, is still at the preclinical stage. Future research on this kind of engineered cells may concentrate on intensifying their capacity for resistance to exhaustion, improving precise positioning and killing effect on target cells, and reversing the immunosuppression of the TME in breast cancer (*Corti et al., 2022*).

Other research has indicated that other components also play an indispensable supplementary role in CAR-T function, including the hinge and transmembrane domains (*Wang et al., 2024*). The subtle relationship between theses various components of the CAR-T system increase its complexity. Even the circadian changes of TME work with some efforts on the recruitment of CAR-T cell (*Chmielewski & Abken, 2015*). The question of how to assemble each component into engineered CD8+ T cells for maximum therapeutic effect in breast cancer still needs further consideration (Fig. 1).

Thus far, the studies of CAR-T for the treatment of solid tumors have led to Phase I/II trials without approval for clinical use. However, the construction of a CAR-T system based on theoretical structures and preclinical research would be of incalculable importance in breast cancer, especially for TNBC, leading to the prospect of applying CAR-T therapy in different directions combined with other treatments. Recently, a breast cancer-on-chip model with an integrated endothelial barrier has been built to evaluate the efficacy and safety of CAR-T cells *in vitro* which can stimulate TME and avoid various limitations *in vivo* (*Maulana et al., 2024*). Meanwhile, we also predict that administering radiotherapy and/or chemotherapy with sequential CAR-T therapy will boost the limited therapeutic effect and improve breast cancer outcomes despite the current lack of clear real-world evidence, particularly in TNBC.

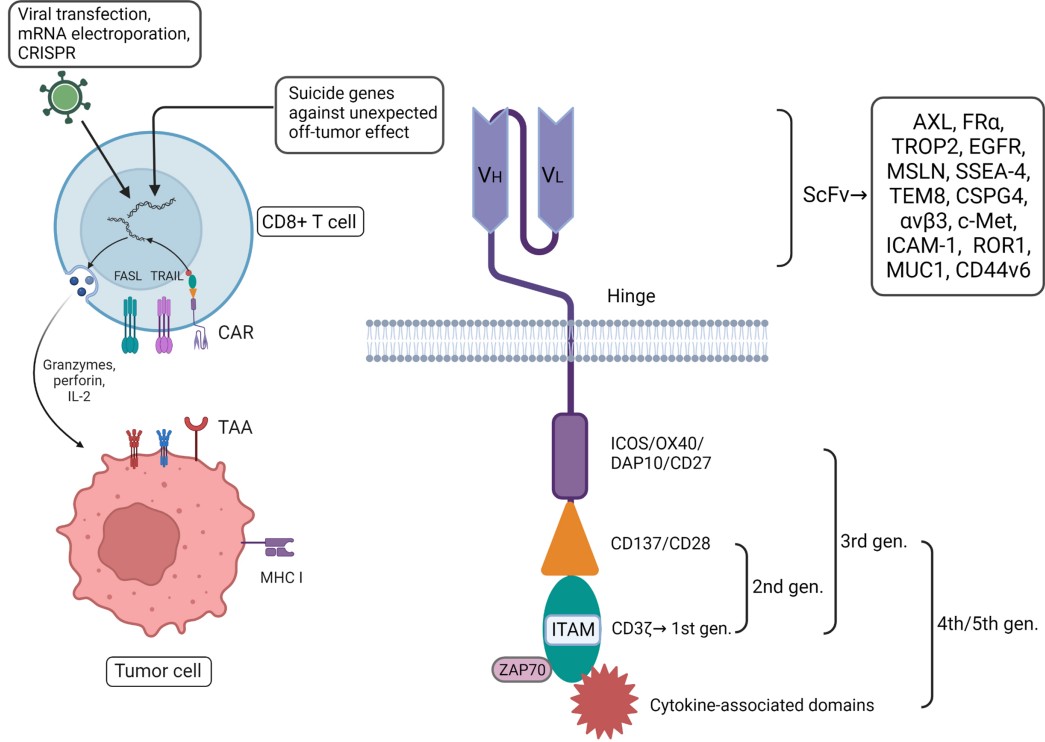

**Figure 1 CAR-T therapy has played a good role in anti-tumor, with more direct and immediate cytotoxicity. CAR-T associated with CD8+ T cells can be programmed with different plug-in components.** Among them, extracellular single-stranded variable fragments (scFv) can be specifically constructed *in vitro* to target breast cancer by contact with tumor-associated antigens (TAAs). The intracellular components are continuously innovated to enhance the comprehensive capabilities of engineered CD8+ T cells. In addition, some suicide genes have been incorporated into the design to avoid detumorization effects. Created with Biorender.com.

Bispecific antibodies (BsAbs) are formed with two fragment antigen binding (Fab) regions, one of which is fixed to target a tumor antigen, while the other either targets a second tumor-associated antigen (TAA)-binding region or recruits CD8+ T cells to attack the tumor following binding with the CD3 molecule of T cells (*Sivaganesh et al., 2021*); and here with the latter expounded emphatically. Unlike the CAR-T system, BsAbs function without cell encoding, from the initial conjugated IgG monoclonal antibody to the heavy-light chain pairing double variable domain immunoglobulin, in which the accuracy of the heavy-light chain pairing can be achieved by the knob-into-hole or CrossMab technique, to the latest bispecific T cell engager (BITE) lack of the Fc portion. However, their salient features of short circulation half-life and potential toxicity lead to clinical restrictions to some extent. VEGFR1, PRLR, FRα, and EGFR are considered specific targets of BsAbs in the MDA-MB-231 TNBC cell line, while CD3-p95HER2-BsAb seems to be promising in the treatment of HER2+ breast cancer (*Sivaganesh et al., 2021*; *Zhou et al., 2020*; *Rius Ruiz et al., 2018*). BsAbs as a separate application still cannot stand up to breast cancer with high heterogeneity, so a combination of CAR-T and BsAbs or several BsAbs may benefit the prognosis of breast cancer. Meanwhile further research is required to ensure their safety.

Rated as an emerging technique with great potential, tumor vaccines have failed to meet expectations thus far. This kind of immunotherapy functions by activating certain related immune cells to fulfill the mission of killing tumor cells. The HER2 protein and peptides derived from HER2 have been studied extensively as tumor epitopes, whereas few vaccines are irrelevant to HER2 (*Zhu & Yu, 2022*). Early trials of tumor vaccines found a visible T cell response or cross-reactive antibody response; however, most late-stage clinical trials have not demonstrated a benefit unambiguously. However, more DFS events have been reported unexpectedly in the presence of the E75 vaccine (*Mittendorf et al., 2019*), which are probably connected with immunosuppressive cells like regulatory T cells (Tregs) driven by the secretion of cytokines induced by the vaccine antigens. GP2, a peptide antigen derived from the transmembrane domain of HER2, is remarkable in that it can prime the specific CD8+ T cell response. It has been already confirmed that the 5-year DFS for human breast cancer cases with HER2 overexpression (3+) treated with postoperative standard targeted therapy combined with this vaccine is 100%. In contrast, TNBC is more likely to benefit from the AE37 vaccine by stimulating CD4+ T cells (*Brown et al., 2020*). Given the limitations of tumor vaccines, such as stored immune tolerance, TME immunosuppression, and tumor heterogeneity, combination therapies including ICBs plus tumor vaccines, CAR-T plus tumor vaccines, and targeted therapy plus tumor vaccines are undoubtedly a breakthrough (*Melief, 2015*). However, the mechanisms underlying the synergy and interaction of these strategies to amplify the immune response need further exploration in the future.

## THE MECHANISMS AND FEATURES OF CD8+ T CELLS IN THE TUMOR IMMUNITY CYCLE

Much research has demonstrated that a high level of CTLs is closely related with clinical benefits, whether through a direct effect on the tumor or indirect assistance with other treatments, thus that is turn out to be winner of anti-tumor process compared with CD4+ T cells (*Melief, 2015*; *Zou et al., 2020*). It is widely known that CTLs expressing the CD8 receptor usually attack tumor cells in the presence of MHC I molecules. In the earliest stage of the immune cycle, DCs capture certain TAAs in their role as potent antigen-presenting cells (APCs) and produce a specific antigen peptide *via* degradation by the proteasome, resulting in a conjugate of the peptide and MHC I molecules that is then transferred from the interior of the cell to the surface (*Rock, Reits & Neefjes, 2016*). The complex can then be recognized by TCRs expressed by CTLs to prime naive CD8+ T cells, which form a wide range of subpopulations including effector T cells (TEFFS) and memory precursor cells. The latter further differentiate into central memory T cells (TCMs), effector memory T cells (TEMs), stem cell memory T cells (TSCMs), and a non-circulating subset of tissue-resident memory T cells (TRMs). Moreover, TCMs, TSCMs, and TRMs seem to be able to self-renew and interconvert (*Zebley, Gottschalk & Youngblood, 2020*; *Han et al., 2020*).

During the initial process of priming CD8+ T cells, CXCL9, CXCL10, CD70, and CD80/ CD86 expressed by DCs interact with receptors expressed by CTLs like CXCR3, CD27, and CD28. The first step in the activation and migration of CD8+ T cells has been found to

occur in the presence of co-stimulating signals, one of which is CD28, evoking a series of cascade reactions following PI3K activation to prolong the survival of T cells. The engagement of extracellular TCRα and β chains and MHC I molecules is involved in downstream signal transduction with the conjugated CD3 complex embedded in cells (*Raskov et al., 2021*). Moreover, the trigger, proliferation, and transfer to the targets of CD8+ T cell are also susceptible to cytokines as IFN-γ secreted by CD4+ T cells and natural killer cells (NKs). Meanwhile, CD4+ T cells can also act directly on DCs or encourage signal transduction between NKs and APCs, which indirectly serve as their own agonists for CTLs (*Topalian et al., 2016*). The evidence mentioned above shows that these subsets of immune cells cannot function separately but communicate with CD8+ T cells in a positive and synergistic way to stimulate them.

The cytotoxic granule formed by synaptic exocytosis is known to be the primary contributor in attacking tumor cells. One mechanism by which this occurs involves the release of granzyme B (GZMB) following the production of perforin 1 (PRF-1) by CTLs, which can induce target cell apoptosis *via* activated caspase 3. In another mechanism, FasL binds with Fas attached to target cells following degranulation and triggers the Fas-associated protein with death domain (FADD), which in turn activates caspase 8 and caspase 3 by a cascade reaction, resulting in toxicity (*Savas et al., 2018*; *Golstein & Griffiths, 2018*). Furthermore, it is found that some cytokines can be also taken out of CTLs, of which IFN-γ and TNF-α tend to activate M1 macrophages, thereby triggering tumor cell death (*Farhood, Najafi & Mortezaee, 2019*). Interestingly, recent evidence suggests that the Notch signaling pathway is relevant to the production of both effectors (*Tsukumo & Yasutomo, 2018*).

The anti-tumor effect of CD8+ T cells on HR– breast cancer with higher immunogenicity is even more striking. Nevertheless exhausting T cells (TEX) is inevitable on account of the persistence of high-level antigens and host self-preservation (*Hashimoto et al., 2018*). Further investigation into the mechanisms of cold tumors like breast cancer, especially TNBC, is essential for the future direction of clinical strategies. Due to the negative influence about high load of tumor antigens and the suppressive TME, intracellular changes involving chromosome aberrations, dysregulated transcription, epigenetic modifications, and metabolic disorders occur, of which the dysfunction of the signaling pathways of diverse cytokines (interleukin (IL)-2, IFN-γ, TNF-α), the overexpression of inhibitory receptors (IRs PD-1, Tim-3, LAG-3), and the reduction of cell lysis, survival, and proliferation are in accordance with the results mentioned above (*He et al., 2019*). It has been reported that cytokines originating from tumor cells and suppressor cells in the TME lead to T cell depletion in different ways; the former contain adenosine, IDO and VEGF-A, while the latter contain IL-10, TGF-β and IL-35. Cancer-associated fibroblasts (CAFs), Foxp3+ Tregs, M2-type tumor-associated macrophages (M2-type TAMs), and myeloid-derived suppressor cells (MDSCs) are the major intermediaries of immunosuppression (*Farhood, Najafi & Mortezaee, 2019*; *Peng et al., 2019*). It is reported that dedifferientiated breast tumor cells can possibly limit CD8+ T cell infiltration to some extent mediated by SOX9-B7x axis (*Liu et al., 2023*). To account for this, the reversal effect of ICBs on TEX has recently gained prominence,

having been particularly noted for PD-1/PD-L1 and CTLA-4 mentioned previously. There is some evidence that PD-1 inhibits immune cells by dephosphorylating the downstream molecules of CD28 and TCR mediated by SHP2, and CTLA-4 expressed by Tregs competes with CD28 embedded in CD8+ T cells for CD80/CD86 from DCs, which tends to decrease the activation of CTLs by DCs (*Farhood, Najafi & Mortezaee, 2019*; *Hashimoto et al., 2018*; *Patel et al., 2021*). Interference with IRs is a recent practical and promising approach for the treatment of TNBC with high immunogenicity even though the capacity for immune escape and higher heterogeneity of breast cancer diminish the reversal effect on TEX. Theoretically, combination therapy is probably superior to monotherapy, with its own sphere of competence. Intriguingly, it has been reported that cytokines like IFN-γ and TNF-α do not always play a helpful or positive role in anti-tumor immunity (*Ou et al., 2018*; *Zhang et al., 2018*; *Huang et al., 2018*). Additionally, the separate interference with cytokines is likely to be of limited value for the modulation of the immune response; in contrast, it is more meaningful and practical to intervene in the upstream pathways of gene mutation and epigenetic modification or the immunosuppressive TME (Fig. 2).

Research has shown that high-level CTLs within cancer islands mediate more efficient protection than CTLs infiltrating the tumor stroma and lead to improved relapse-free survival (RFS), especially in cases of TNBC and HER2+ breast cancer. Furthermore, the enrichment of CD8+ T cells in the tumor stroma has been exhibited though quantitative immunofluorescence (QIF), whereas differentiated TRMs are generally present in cancer islands and epithelial tissues (*Peng et al., 2019*; *Egelston et al., 2019*). The mechanism of suppression relying on tumor to CTLs infiltration is still unclear, and the significance of TRMs will be illustrated below in the immune response to breast cancer. As one of the suppressor cell types against CTLs, the distribution and density of Foxp3+ Tregs agrees with those of CD8+ T cells simultaneously (*Peng et al., 2019*). The feature of tolerance for immune surveillance is the main challenge in the treatment of a cold cancer like breast cancer, and the effect of treatment associated with CD8+ T cells would be subject to this feature. It is supposed that CTLs can be applied in the treatment of breast cancer in cooperation with targeted inhibitors of immunosuppressive cells like Tregs, which play a central role in anti-tumor immunity, and that this strategy probably has a magnifying effect on CD8+ T cells, contributing to higher efficiency (*Saleh & Elkord, 2020a*, *2020b*).

The first port in breast cancer metastasis is generally the sentinel lymph nodes (SLNs); in this regard, breast cancer is distinct from other solid tumors. Recently, it was reported that almost 60% of CD8+ T cells could be recruited to the SLNs. Unexpectedly, the gene expression of CTLs in metastatic SLNs is not exactly the same as in the primary tumor tissue; even their number can produce genomic differences, likely related to the transcription signals of specific homing and antigen processing steps (*Liao et al., 2023*; *Chen et al., 2021*). This mechanism has not yet been confirmed, but it may be related to the interaction between immune cells. In a sense, the enrichment of CTLs in the SLNs is conducive to immunotherapy but may be a symbol of higher heterogeneity in metastatic breast cancer. The present challenge is to target and boost the immune response of CD8+ T cells in lymph node (LN) regions due to the potential significance of LN immune status in the assessment of breast cancer prognosis. Recent research showed that nanoparticles

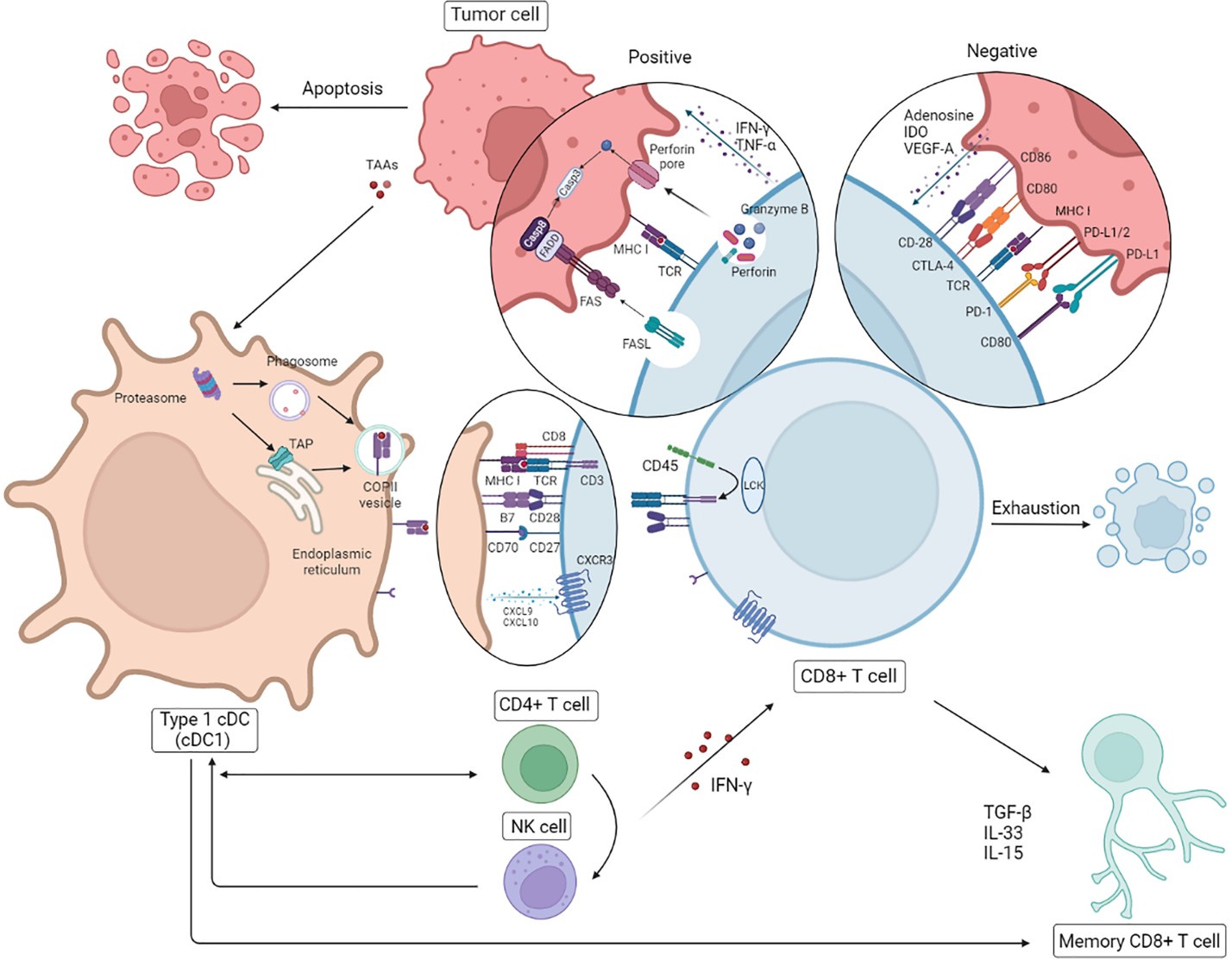

**Figure 2 Correlation between CD8+T cell and other immune cells.** Following presentation of tumor antigens by specific antigen-presenting cells (APCs), CD8+ T cells produce an immune response to tumor cells involved in cytokines. In breast cancer, in addition to the positive anti-tumor effects, there are some cases of T cell failure caused by a negative immunosuppressive process mediated by tumor cells through immune checkpoints. NK, natural killer; DC, dendritic cells. Created with Biorender.com.

functioning as specific medicine carriers could target SLNs and activate T cells locally (*Li & Hung, 2022*). This finding suggests that therapies for metastatic breast cancer related to CD8+ T cells can be applied under hierarchical control with multiple targets to support precision medicine in the future.

# FACTORS INFLUENCING CD8+ T CELLS

## The TME

As mentioned above, cancer tolerance and immune escape can be attributed to various immunosuppressive components in the TME that involve several molecular pathways directly or indirectly. Apart from the secretion of negative factors like adenosine and

TGF-β caused by synergy with Tregs, tumor cells protect themselves against the activity of CD8+ T cells through the expression of immune checkpoint proteins, such as PD-L1/PD-L2, B7.1, B7.2, and ICOS-L (*Raskov et al., 2021*). *Gruosso et al. (2019)* delineated that there was some differences between the various time of TME related to the spatial distribution and density of CD8+ T cell corresponding to different subtypes ofTNBC prognosis. M2-type TAMs have recently been shown to achieve the suppression of CTLs by increasing IL-10 and IDO levels, in addition to mutual benefit with breast cancer cells. The prognostic value of M2-type TAMs in breast cancer can be realized by monitoring CD163, ARCO, CD206, CD204, and other markers (*Qiu et al., 2018*). Furthermore, NKs, CD4+ T cells, DCs, and M1-type TAMs establish a positive interaction with CD8+ T cells, in contrast with suppressor cells like Tregs, MDSCs, and CAFs. As one of the important effectors of CTLs, TNF-α also appears to have a negative effect through the regulation of the TME, possibly with TGF-β (*Farhood, Najafi & Mortezaee, 2019*; *Ou et al., 2018*). Likewise, the positive effect of INF-γ on CTLs may be reduced with continued exposure. For instance, the beneficial effect of INF-γ in inducing differentiation into memory cells might be converted into the deleterious effect of mediating TEX *via* PD-L1 (*Alspach, Lussier & Schreiber, 2019*). Moreover, there is a close affinity between ILs and the development of CTLs. IL-12, IL-2, IL-10, and IL-15 are positively correlated with CTLs in breast cancer, whereas IL-6 and IL-33 have the opposite relationship (*Saraiva, Vieira & O'Garra, 2020*; *Fiore et al., 2020*; *Shen, Liu & Zhang, 2018*). Considering their close links with other immune components, cytokines can be used as immunological adjuvants but require further exploration to be applied in the clinic. It has been pointed out that STAT3 signaling, which is strongly involved in the immune response, is not necessarily prohibitive for CD8+ T cells, depending on the differentiation of T cells (*Verdeil et al., 2019*). No component can act alone in the tumor immune cycle, and the underlying multifactorial mechanisms remain controversial. Whether these effectors act as agonists or inhibitors seems to depend on whether the expression of each component alone and together with other factors is suitable for the development of CTLs.

## Hormones

Notably, the level of infiltrating CD8+ T cells in HR+ breast cancer is lower than in HR– breast cancer. Currently, research is being carried out to analyze the effect of CTLs on HR+ breast cancer, in which CTL infiltration is positively associated with response to chemotherapy, an effect that is probably linked to the mTOR signaling pathway (*Schettini et al., 2020*). However, the relationship between hormones and CTL levels and the associated mechanisms are rarely discussed. For example, estradiol affects CTLs by helping to regulate pro-inflammatory (IL-2, IL-12, IFN-γ, TNFα) and anti-inflammatory (IL-10, IL-4, TGF-β) cytokines (*Segovia-Mendoza & Morales-Montor, 2019*). Additionally, estrogen is thought to dampen the activity of CTLs through the granzyme pathway (*Schettini et al., 2020*). In the SOLTI-1501 VENTANA trial, infiltrating lymphocytes in HR+ breast cancer were elevated upon endocrine treatment compared to treatment with the placebo (*Adamo et al., 2019*), and we presume that hormones might be somehow counterproductive to CD8+ T cells. Yet, the thought of raising CTL levels to improve the

prognosis of luminal breast cancer with unclear benefits is not in accordance with the conventional clinical treatment regimens for HR+ breast cancer (*Terranova-Barberio et al., 2020*), which is based on further research on relevant signaling pathways and mechanisms.

### Human intervention

Breast cancer one of the best known cold tumors, and in this regard, the importance of combination therapy is beyond question. A deep understanding of the connection between therapies and the level of infiltrating CTLs is required for the formulation of appropriate clinical strategies. Conventional chemotherapy may hinder the immune response due to its toxicity. Interestingly, it has been reported that the evolution of TCRs bound to CTLs occurs during chemotherapy and is associated with clinical efficiency even though this evolution is taking place without orientation and uniformity (*Lin et al., 2018*). Thus, chemotherapy cannot be characterized simply as an immune suppressor. Interestingly, CXCL13+ CD8+ T cell level in TNBC patients with the combination therapy of paclitaxel and atezolizumab was seemingly higher than those with paclitaxel alone (*Zhang et al., 2021*). Similarly, radiotherapy has two sides that can prime CD8+ T cells *via* APCs as well as recruit and upregulate immunosuppressive components. The specific correlation between radiation sensitivity and radiation-induced lymphocyte apoptosis (RILA) is still unclear (*David et al., 2022*; *Fhoghlú & Barrett, 2019*). Compared with radiotherapy, precise ablation, a new treatment for breast cancer, plays a prominent role in potentiating the immune system. More and more evidence shows that microwave ablation (MWA) helps to release high amounts of TAAs and cytokines to reshape the TME apart from precisely attacking the local tumor, which is possibly related to the activation and proliferation of CD8+ T cell (*Xiao, Wu & Jiang, 2024*). However, some research demonstrates that CD8+ T cells activated by MWA have impaired immune features, such as TEX and emission dysfunction of IFN-γ (*Zhou et al., 2022*). Thus, more evidence is required regarding the paradoxical effects of ablation on the immune system. It has been suggested that the number of CTLs and PD1+ CD8+ T cells increases in non-PCR breast cancer with neoadjuvant chemotherapy (NACT); additionally, the significance of the engagement of PD-1 inhibitors following chemotherapy for non-PCR breast cancer patients has been stated (*Liang et al., 2020*). NACT causes the formation of a pro-inflammatory environment and mediates the cytotoxicity and activity of CTLs *via* IFN-γ and IL-12 (*Osuna-Gómez et al., 2021*). CTLs and CD4+ T cells rise synergistically in postmenopausal patients with advanced HR+ breast cancer who benefit from the mTOR inhibitor everolimus (*Schettini et al., 2020*). All these findings indicate that different therapies affect CD8+ T cells to various extents while confirming the positive relationship between the level of CTLs and response and prognosis. Presently, research is concentrated on achieving maximum clinical benefit by adding immunotherapy to individual therapy.

## DEFINITION AND CHARACTERISTICS OF CD8+ TRMS IN BREAST CANCER

Most naive CD8+ T cells differentiate into effector cells to function as local immune supervisors and killers in tumors, and exhaustion occurs in 95% of this population within a

short period, in view of the signaling pathways mentioned above (*Zebley, Gottschalk & Youngblood, 2020*). Introduced as a specific cell lineage developing from CD8+ T cells, TRMs play a vital role in the maintenance of the anti-tumor immune response in cancers with higher heterogeneity, especially breast cancer originating from the epithelium. It is also suggested that high infiltration of TRMs indicates a favorable prognosis (*Molodtsov & Turk, 2018*). So far, biological markers for this specific subpopulation of CD8+ T cells mainly include CD103 (αEβ7), CD69, CD49a (VLA-1), and PD-1 (*Mami-Chouaib et al., 2018*; *Kumar et al., 2017*). A study showed that CD103 and CD49a could enhance the maintenance of TRMs in conjunction with E-cadherin and collagen IV associated with tumor cells where cytotoxicity could be further developed. CD69 acts as a marker for T cell activation, which could interfere with the expression of tumor promoters like S1PR1 and CCR7 (*Mackay et al., 2015*). Apart from counteracting the egress of TRMs through the upregulation of upstream regions and signaling molecules for maintenance and adhesion, TRMs can also strengthen peripheral tolerance *via* various pathways, such as HAVCR2, CTLA4, PDCD1, LAG3, and TIGIT (*Byrne et al., 2020*; *Corgnac et al., 2018*; *Han et al., 2020*). Intriguingly, most of these genes are related to immune checkpoint proteins like PD-1, CTLA-4, and Tim-3 (*Mami-Chouaib et al., 2018*) that mediate TEX as mentioned previously; therefore, further research should be conducted on the partnership between ICBs and the viability of TRMs. Moreover, there are certain special transcription factors in TRMs, including KLF2, Notch-RBPJ, RunX3, Hobit, BATF, Blimp1, Enome and Tbet (*Mami-Chouaib et al., 2018*; *Milner et al., 2017*; *Mackay et al., 2016*). Tbet is repressed by TGF-β in the development of TRMs and negatively correlates with memory cell survival (*Kallies & Good-Jacobson, 2017*; *Barros, Ferreira & Veldhoen, 2022*). However, there's evidence that these markers are not necessarily germane to TRMs, and TRMs have different expression in different tumors (*Wang et al., 2020*); thus, TRMs need to be identified using multiple targets. The main recognition sites of TRMs in breast cancer are CD103 and CD69, corresponding to unique genetic features, including the upregulation of ITGAE, ITGA1, CD244 and XCL1 and the downregulation of S1PR1 (*Egelston et al., 2019*).

Regarding TEX metabolism, this process results from the competition between CD8+ T cells and tumor cells for glucose and oxygen. In contrast to the general metabolism of CD8+ T cells, TRMs can strengthen the utilization of free fatty acids (FFAs), responsible for longer survival (*Pan & Kupper, 2018*). Recent research has found that FABP4 and FABP5 in TRMs manage this process (*Byrne et al., 2020*); however, it is unclear whether there are differences between various types of tumors. The metabolism of TRMs in breast cancer needs to be confirmed for the future development of adjuvants related to TRMs.

It is a notion that DNGR-1, a C-type lectin receptor (CLR) expressed by type 1 cDCs (cDC1s) in humans, induces CD8+ T cells to differentiate into TRMs with TGF-β, IL-33, and IL-15 through the downregulation of the expression of CD62, Dnmt3a and KLRG1 (*Mami-Chouaib et al., 2018*; *Barros, Ferreira & Veldhoen, 2022*; *Cueto, Del Fresno & Sancho, 2020*; *Youngblood et al., 2017*; *Craig et al., 2020*). Following differentiation, the migration and invasion of T cells is mediated by CXCR3 and CCR9 independently

(*Yang & Kallies, 2021*). It has been suggested that TRMs, as terminally differentiated non-circulating effector cells, have a certain plasticity and proliferative ability that does not require the assistance of CD4+ T cells (*Beura et al., 2018*; *der Gracht ETI, Behr & Arens, 2021*). TRMs can recruit circulating non-specific memory T cells (TMs), and bystander CD8+ T cells have the potential to differentiate into TRMs with or without the stimulation of local antigens, although cellular location and proliferation are restricted by a lack of CD103 (*Beura et al., 2018*; *Behr et al., 2021*). Furthermore, local inflammation is involved in the development of TRMs (*der Gracht ETI, Behr & Arens, 2021*). Based on cell detection *in vitro*, CD8+ TRMs in breast cancer have been unexpectedly found to express more cytotoxic granules like GZMB, PRF1, CCL5, IFN-γ, and the degranulation marker LAMP-1 (CD107a) than common CD8+ T cells (*Mami-Chouaib et al., 2018*), which may be connected with the reconstruction of the cellular synaptic configuration following the combination of CD103 and E-cadherin (*Molodtsov & Turk, 2018*). However, it is not certain that the strong ability of CD8+ TRMs to kill tumor cells is bound to cytotoxicity. Recently, it was proposed that the extraordinary effect of CD8+ TRMs on breast cancer has some connection to localization. The density of TRMs in cancer islands is more significantly related to RFS by downregulating S1PR1 and upregulating ITGA, CD244, and XCL1 for keeping position. Additionally, the tropism of TRMs for cancer islands is clearly more beneficial for the prognosis of breast cancer than that of other subsets of CD8+ T cells with similar functions (*Egelston et al., 2019*). We believe that whether the quantity or quality of TRMs gains an advantage depends on the competition within different types of tumors. Thus, the predominant targets for clinical applications should be in accordance with the characteristics of different tumors and the corresponding TRMs, as well as their interactions.

## APPLICATION AND PROSPECT OF TRM IN BREAST CANCER

The anti-tumor effect of different subtypes of TRMs corresponding to various solid tumors has been confirmed in mouse models of cancer, including melanoma, non-small cell lung cancer (NSCLC), and ovarian cancer. Recently, *Savas et al. (2018)* revealed that CD8+ TRM s were intimately related to a favorable prognosis in TNBC based on the single-cell RNA sequencing (sc-RNA-seq) of T cells isolated from breast cancer tissue. However, a recent research had demonstrated that CD103 is positively connected with the better prognosis in luminal breast cancer (*Chan et al., 2024*). As mentioned above, there is no explicit relationship between the spatial distribution and mechanisms of TRMs and the genetic heterogeneity in mice and humans. Apart from predicting survival, TRM-associated therapies have not been applied in real-world settings, and research progress is limited regarding breast cancer. Nonetheless, the potential applications of TRMs cannot be ignored and underestimated. The next steps in exploring the applications of TRMs will be based on developments in increasing the quantity of TRMs and their efficiency as tumor suppressors (Fig. 3).

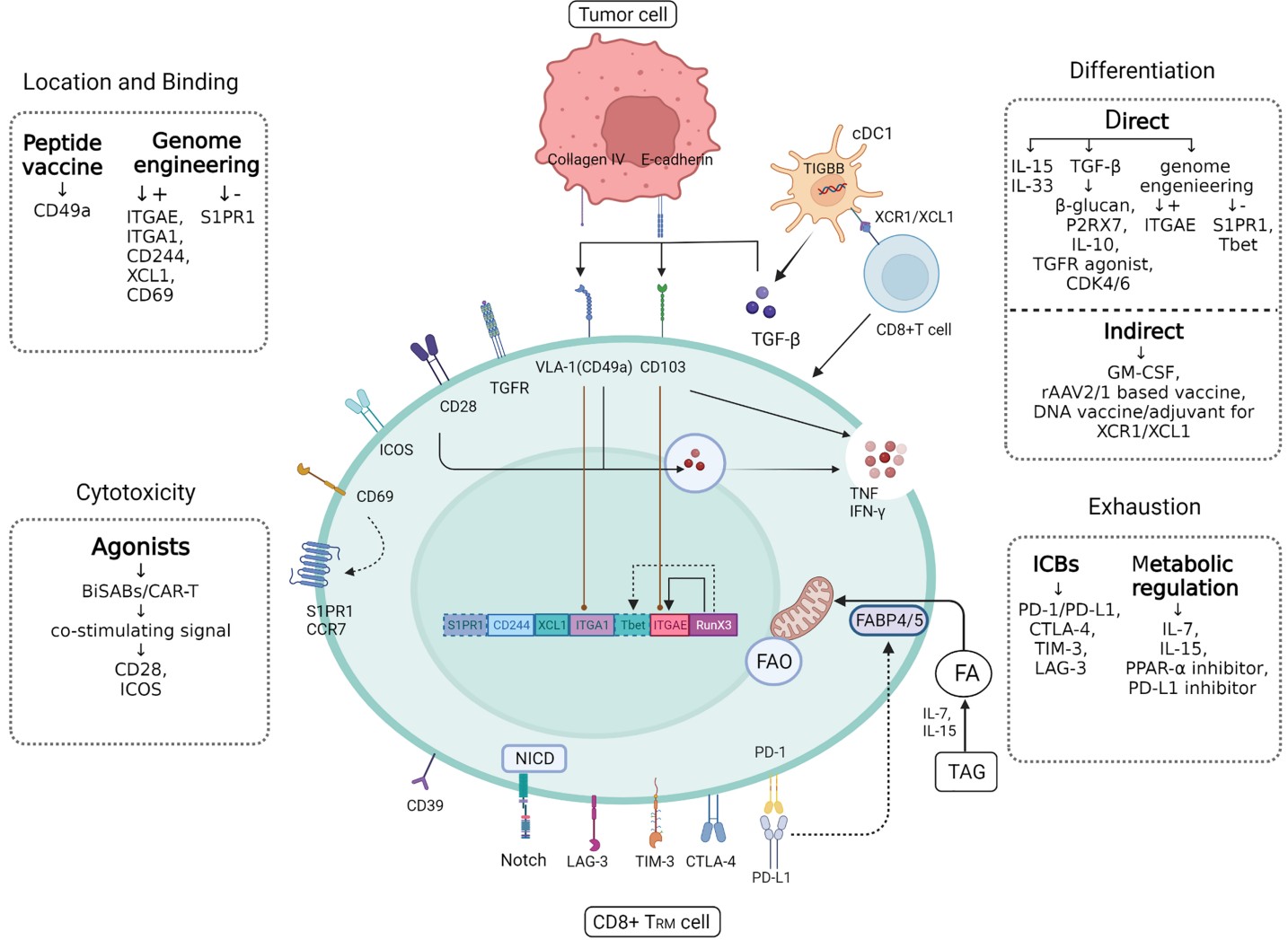

**Figure 3 CD8+ TRM cell.** Figure 3 CD8+ TRM has a more powerful and rapid attack on tumor cells than ordinary CD8+ T cells, and future applications in breast cancer should focus on how to target to increase the efficiency and quantity of TRM as a tumor suppressor. In addition to the various targets of TRM synergistic localization, adhesion and killing of tumor cells, there are also some significant cytokines that can directly or indirectly induce CD8+ T cells to differentiate into CD8+ TRM, such as TGF-β produced by cDC1. In addition, immune checkpoints often mediate self-monitoring and management of TRM, which may be interfered with by immune checkpoint blockers (ICBs) and metabolic regulation to better utilize fatty acids (fas) against tumor cells. Created with Biorender.com.

## Enhancement of TRM
### *Resisting against TEX*

As a member of the CD8+ T cell group, CD8+ TRMs share the properties of self-surveillance and management mediated by immune checkpoints like PD-1 to protect the host, which result in some drawbacks regarding anti-tumor immunity. *Edwards et al. (2018)* reported that the application of anti-PD-1 and/or anti-LAG-3 could probably enhance the positive effects of TRMs on the prognosis of melanoma. Predictably, TRM self-resistance can be reduced from the supplement supported by ICBs properly, which has a pattern similar to that of Notch signaling inhibitors (*Zhang et al., 2017*). However, Notch signaling sometimes appears to be common to both CD8+ TRMs and breast cancer cells;
thus, it is necessary to identify non-overlapping targets in the signaling pathway for optimal interventions. More pre-clinical and clinical evidence is required to achieve a delicate balance between the restriction and activation of TRMs in humans.

Recently, it has been demonstrated that the expression of FABP4/FABP5 can be promoted by anti-PD-1 therapy *via* another pathway along with the lipid dysbolism of tumor cells genetically (*Lin et al., 2020*). TRMs have a competitive advantage in using FA for energy metabolism over tumor cells, which rely heavily on glucose metabolism. Furthermore, drugs can be custom-designed to target TRM metabolic pathways. It has been shown that IL-7 and IL-15 synergistically drive triglyceride (TAG) synthesis and lipolysis in memory CD8+ T cells to generate materials from which FAs originate (*Cui et al., 2015*). Therefore, adjuvants like an IL-7 recombinant vaccine or PPAR-α agonist show promise in boosting TRM survival in breast cancer (*Zhang et al., 2017*). Unfortunately, fatty acid oxidation (FAO) prevails in TNBC metabolism (*Zhang, Xu & Ye, 2021*), so that the use of fatty acids by TNBC cells creates an unexpected barrier to direct interventions based on TRM metabolism.

### Induction of CD8+ T cell differentiation into TRMs

Increasing the number of TRMs seems to be more practical than intervening against TEX during whole-cell differentiation. Regarding the priming of TRMs, it is known that TGF-β and integrins like αvβ8 derived from cDC1s are key components contributing to the further differentiation of CD8+ T cells (*Wu et al., 2014*). There is no simple one-way links among all of them, that CD8+ T cells can also be influenced by antigen-specific TRMs with the media of DCs in the reverse direction (*Menares et al., 2019*). Intriguingly, a recent trial *in vitro* revealed that DC3, a subunit of cDC1s, significantly induces CD8+ T cell differentiation into TRMs, an effect that can be strengthened by granulocyte-macrophage colony-stimulating factor (GM-CSF) (*Bourdely et al., 2020*). Moreover, CD8+ T cell activation in breast cancer is in accordance with the upregulation of ITGB8 expression in DCs by β-glucan (*Blanc et al., 2018*). Inputting reprogrammed DCs and/or some related stimulating factors may be a potential method. It has been reported that a recombinant adeno-associated virus (rAAV) vector vaccine and XCL1 vaccine adjuvant assist APCs in presenting antigens, with indispensable cross-presentation driven by CD4+ T cells (*Gross et al., 2019*; *Matsuo et al., 2018*). In conclusion, the simultaneous enhancement of presentation mediated by different APCs probably enables TRM activation to a great extent. Though TRMs are likely to proliferate autonomously, it has been suggested that signals from specific antigens are indispensable for the formation of fully functional, productive TRMs (*Muschaweckh et al., 2016*). Due to the potential cross-competition of TRMs, antigenic epitopes are meant to be reprogrammed in tumor vaccines to achieve more specific TRMs with a higher killing efficiency.

Theoretically, stimulating TRMs *via* specific targets tends to be much more direct and selective. It is known that TGF-β and IL-15 play an indispensable role in the activation of TRMs, wherein the purinergic receptor P2RX7 fosters the response to TGF-β (*da Silva et al., 2020*). It is conducive for TGF-β to be secreted in combination with latency-associated protein(LAP), following being activated by metalloproteinases and

αvβ6 and αvβ8 integrins (*Damei et al., 2023*). The corresponding target is available for application in the future. The members of the TNFR superfamily include 4-1BB, OX40 and GITR, agonists of which would serve to regulate the maintenance of TRMs (*Zhou, Batista & Watts, 2019*; *Zappasodi et al., 2019*; *Pritzl, Daniels & Teixeiro, 2021*). IL-10 also plays an assistive role in the secretion of TGF-β. *Mackay et al. (2015)* confirmed that TRM survival has a certain dependence on IL-15 that is possibly associated with the inhibition of mTOR (*Heckler et al., 2021*). Interestingly, CD4/6, which affects tumor cell attack *via* cell cycle regulation, can also establish a memory CD8+ T cell pool in the early immune response to breast cancer *via* the MYC/MXD4 axis and skew CD8+ T cells toward TRMs (*Heckler et al., 2021*). However, a Phase Ib study suggested that a CDK4/6 inhibitor could unexpectedly lead to serious liver and lung damage when combined with a PD-1 blocker in NSCLC and HR+ HER2– metastatic breast cancer (*Pujol et al., 2021*). Thus, the partnerships between different adjuvants for TRMs need to be studied further and coordinated. Various chemokines derived from CD8+ TRMs in different tissues modulate cell growth and proliferation as well as the targeting ability of TRMs. For instance, CXCR3, CXCL9, and CXCL10 are applied to the skin and female reproductive tract (FRT) and are conducive to the formation of TRMs (*der Gracht ETI, Behr & Arens, 2021*). Thus far, there is little focus on tracing the role of chemokines as dominant players in breast cancer. Moreover, the transcription factors mentioned above are somehow positively or negatively related to the establishment of TRMs. Among them, RunX3 has an early and significantly positive effect on T-bet inhibition, referred to as a negative regulator, and ITGAE (CD103) enhancement, apart from an independent motivation for TRMs (*Milner et al., 2017*). More evidence is required to verify whether these genes can be reprogrammed to achieve the long-term maintenance of TRMs. If relevant cytokines can be used in combination with CTL-associated therapies, they should be given high priority in immunotherapy.

## Strategies to boost the anti-tumor efficiency of TRMs

It is supposed that interventions aimed at realizing the efficiency and potential of TRMs should focus on targeting and cytotoxicity separately. Based on the functional analysis of the integrins mentioned above, CD69/CD49a and CD103 are largely responsible for the retention of CD8+ TRMs. Nevertheless, it's proved that the upregulation of CD39 and Tim-3 is contributed to the ability of residence which is considered as a dynamic procedure, instead of CD69 and CD103 (*Gavil et al., 2023*). In pre-clinical trials of breast cancer, the expression of TGF-β could be amplified so that adhesive molecules were effective against tumor cells (*der Gracht ETI, Behr & Arens, 2021*). Peptide vaccines help CD8+ T cells to express CD49a in melanoma, and CD49a later induces cytotoxic granules like IFN-γ (*Melssen et al., 2018*). It is known that some transcription factors are location-dependent of TRMs in breast cancer primarily include S1PR1, CD244, and XCL1 (*Egelston et al., 2019*). Therefore, cytokines and/or related antibodies can be used to modulate the directional viscosity and tropism of TRMs *via* the stimulation and protection of these integrins and cellular receptors, which is an optimistic but not completely far-fetched vision. Meanwhile, co-stimulating signal molecules, like CD28 and ICOS, are

the perfect addition to reinforce the production of TNF and/or IFN-γ, the latter of which seems to be especially involved in the early differentiation of TRMs (*Peng et al., 2022*). Compared with chemotherapy alone, TRM positive impact on the prognosis of TNBC with the combination of PD-L1 and chemotherapy is more significantly statistically, mainly because of its high quantity (*Virassamy et al., 2023*).

T cell differentiation need not be constant and beneficial *in vivo*. Based on this view, we assume that specific CD8+ TRMs can be programmed genetically, multiplied *in vitro*, and applied in later infusions to maximize their function of attacking tumor cells, which is a form of ACT (mentioned above). Though the results of *in vitro* experiments may seem less than ideal (*Behr et al., 2021*; *Komdeur et al., 2016*), the idea provides a way to replenish TRMs following exhaustion for application in anti-tumor treatments. The ongoing immune response to cold breast cancer may be unpredictable; therefore, cell culture methods should be fully considered to coordinate with multiple factors or targets to obtain differentiated TRMs.

## CONCLUSION AND PROSPECTS

As a type of cold tumor with high heterogeneity, breast cancer is usually treated with conventional chemotherapy, radiotherapy, and endocrine therapy with the assistance of molecular targeted drugs, which enable the precise attack of tumor cells, resulting in a favorable prognosis. However, TNBC has no precise and stable targets unlike other subtypes of breast cancer, limiting the clinical application of immune therapy and resulting in a poor prognosis. Currently, PD-1/PD-L1 antibodies, including nivolumab, pembrolizumab, and atezolizumab have been approved by the FDA as part of the combination regimen for metastatic TNBC (*Zhu et al., 2020*). However, clinical evidence suggests that the positivity rate of PD-1/PD-L1 expression by tumor cells and immune cells in TNBC is only 38.7% and 32.2%, respectively (*Sugie et al., 2020*); thus, it is necessary to discuss other immune pathways for the function of CD8+ T cells, as the dominant subpopulation in TNBC. In this review, we summarize the potential targets and application prospects of cellular programs related to CD8+ T cells in breast cancer. Apart from for BsAbs and tumor vaccines, CAR-T systems in connection with CD8+ T cells, regarded as one kind of ACT, expand the landscape of TNBC treatment, with a larger number of available editable targets and a more direct killing effect on tumor cells. We also stress that the combination strategy is promising and necessary for breast cancer with high heterogeneity. However, treatment with multiple targets does not always indicate absolute clinical benefit as toxic effects on non-tumor tissues cannot be ignored (*Corti, Nicolò & Curigliano, 2021*; *Shah et al., 2023*). The impact of the TME, hormone levels, and different interventions on CTL infiltration have also been investigated, and interactions between various factors should be considered and balanced comprehensively when breast cancer is treated with a combination of adjuvants associated with CTLs.

The CD8+ T cell subpopulation of TRMs has been confirmed to play a crucial role in immune surveillance and the immune response in breast cancer relative to general CD8+ T cells characterized by short longevity, slow action, and poor targeting and attacking

abilities (*Savas et al., 2018*). Regarding CTL-associated therapy in breast cancer, a few reviews address two aspects of this subject, namely how to improve the quality and the quantity of TRMs. We also summarize and extend the analysis of the developmental properties of CD8+ TRMs combined with their potential targets. Differences in TRM phenotypes and expression features have been found in different types of tumors (*Wang et al., 2020*), which contribute to the uniform specific treatment related to TRM. Thus far, people have even more constrained knowledge about TRMs, especially in relation to breast cancer. Apart from predicting survival, whether the location or cytotoxicity of TRMs has a significant effect on prognosis is currently a matter of dispute (*Egelston et al., 2019*; *der Gracht ETI, Behr & Arens, 2021*), and CTL-associated therapy has not been applied in the real world. The prospects of combination therapy directly assisted by TRMs oriented in breast cancer, particularly in TNBC, are unclear. Apart from exploring ways to increase the quantity and quality of CD8+ TRMs in breast cancer, approaches to balance the positive and negative effects of T cells in the human body and maximize their clinical benefit warrant exploration in the future.

### Funding

This work was supported by the 2023 Liaoning Science and Technology Joint Plan (2023-BSBA-100) and the 2023 Dalian Medical Science Research Project (2312012). The funders had no role in study design, data collection and analysis, decision to publish, or preparation of the manuscript.

### Grant Disclosures

The following grant information was disclosed by the authors:
2023 Liaoning Science and Technology Joint Plan: 2023-BSBA-100.
2023 Dalian Medical Science Research Project: 2312012.

### Competing Interests

The authors declare that they have no competing interests.

### Author Contributions

- Ziqi Zhao analyzed the data, authored or reviewed drafts of the article, and approved the final draft.
- Xinyu Ma performed the experiments, analyzed the data, prepared figures and/or tables, authored or reviewed drafts of the article, and approved the final draft.
- Zhengang Cai conceived and designed the experiments, authored or reviewed drafts of the article, and approved the final draft.

### Data Availability

   This is a literature review.

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
