# Peer review of "The potential role of CD8+ cytotoxic T lymphocytes and one branch connected with tissue-resident memory in non-luminal breast cancer"

_PeerJ, doi:10.7717/peerj.17667_

## Round 0.1 · original submission · Minor Revisions

The authors are requested to carefully revise the manuscript and answer the questions raised by the reviewers.

**Language Note:** The review process has identified that the English language must be improved. PeerJ can provide language editing services - please contact us at [email protected] for pricing (be sure to provide your manuscript number and title). Alternatively, you should make your own arrangements to improve the language quality and provide details in your response letter. – PeerJ Staff

Reviewer 1 ·

Basic reporting

This literature review on CD8+ T cells and TRM and breast cancer provides a comprehensive overview of the topic. It effectively captures recent research findings about the CD8+ T cells in breast cancer and provides valuable insights for future directions. However, the organization of each section needs to be more structured and clearer for readability. Overall, it is a solid review, but some refinements could further enhance its validity.

Experimental design

The methodology appears to be consistent and recent research developments in this field are captured. The flow is well-structured. However, there is room for improvement in the organization of each section to enhance readability and clarity, especially for section 1 and 2. For example, section 1 discussed different immunotherapies, and subsections could be introduced to delineate distinct aspects of immunotherapy discussed. Meanwhile, the figures provide a clear overview of key concepts such as components of CART and mechanisms of CD8+ and TRM cells. But when these figures are mentioned in the context, they are not properly incorporated. There is a need for better integration of these figures within the text to aid comprehension and reinforce the organization of different sections.

Validity of the findings

The introduction successfully highlights the importance of examining the role of CTLs in breast cancer, setting the stage for the subsequent sections. The following sections addressed this central goal, and the conclusion proposed several future directions about immunotherapy development. However, both abstract and introduction emphasized the importance of looking into TNBC. However, throughout the manuscript, adequate discussion on this topic is lacking. Expanding the discussion in the mechanisms specific in TNBC is suggested.

Additional comments

Improvements in language are necessary to address grammatical, spelling, and formatting errors present throughout the manuscript. For example, section 1, “In addition, in addition to passive …”, “CAR-cars”. Additionally there are several instances of unfinished sentences, for example, section 2, “While in the other mechanism, FasL binds to FAS attached to target cells”, “In particular, TNBC and HER(+).”. Lastly, the absence of a subtitle for section 5.1 ("Improving the Quantity") should be added to maintain consistency in section numbering.

Reviewer 2 ·

Basic reporting

The manuscript by Ma et al. provides a comprehensive overview of the role of CD8+ T cells and tissue resident memory T cells in non-luminal breast cancer. Overall, the authors provided a depth of discussion on the MOA and therapeutic application of CD8+ T cells and TRMs in breast cancer.

However, the linguistic quality of the manuscript should be significantly improved to ensure that an international audience can clearly understand the text. There are multiple typos, grammatical errors, and inconsistencies that impair the clarity and professional presentation of the research, especially for the capitalization and the abbreviation. For example, nivolumab was miswritten as navomumab in the manuscript, dendritic cells are referred as both DC and dc, bispecific antibodies are referred as BsAbs and BiSABS, single chain fragment variables are referred as scFvs and SCFVS, the plural form of TRM should be TRMs instead of TRMS, Tex usually refers to exhausting T cells instead of T cell exhaustion. It's highly recommended that the manuscript undergoes a thorough proofreading by a fluent English speaker or professional editing service to avoid ambiguous language.

Experimental design

Minor comments:

1) In the introduction, the authors used the magnitude of T cells from a previous review of 13,914 breast cancers to indicate the potential of higher immunogenicity of TNBC and HER2+ breast cancer. In general, the density and localization of T cells are more relevant to the therapeutic efficacy (with immune cold/hot microenvironment), and immunogenicity is also associated with the antigen presentation and mutational burden of cancer cells.

2) The manuscript would benefit from a deeper investigation into more relevant studies. For example, Gruosso et al. (PMID: 30753167) demonstrated the subtypes of tumor immune microenvironment in TNBC based on spatial resolution of CD8+ T cell infiltration and gene expression profiles; Liu et al. (PMID: 37963469) showed CD8+ T cell infiltration in the microenvironment was limited by a SOX9-B7x axis to promote TNBC progression; Zhang et al. (PMID: 34653365) revealed that CXCL13+ T cell levels can predict the responses to the combination therapy of paclitaxel and atezolizumab in TNBC patients.

In addition, some references in the bispecific antibody and CAR-T sections are relatively out of date, it's recommended to replace with the most updated research.

Validity of the findings

The arguments in the manuscript are well supported by the findings cited.

---

## Round 0.2 · accepted · Accept

After revisions, all reviewers agreed to publish the manuscript. I also reviewed the manuscript and found no obvious risks to publication. Therefore, I also approved the publication of this manuscript.

Reviewer 1 ·

Basic reporting

All my comments have been implemented in the revised version of the manuscript.

Experimental design

All my comments have been implemented in the revised version of the manuscript.

Validity of the findings

All my comments have been implemented in the revised version of the manuscript.

Reviewer 2 ·

Basic reporting

The authors have addressed all my questions and concerns. The review is acceptable for publication after correcting a few typos:

Line 142: these
Line 243: dedifferentiated
Line 252: diminishes
Line 300: "ofTNBC", missing a blank space

Experimental design

The authors have addressed all my questions and concerns

Validity of the findings

The authors have addressed all my questions and concerns